DATA RELEASE

# Annotation of segmentation pathway genes in the Asian citrus psyllid, *Diaphorina citri*

Sherry Miller[1,2], Teresa D. Shippy[1], Prashant S. Hosmani[3], Mirella Flores-Gonzalez[3], Lukas A. Mueller[3], Wayne B. Hunter[4], Susan J. Brown[1], Tom D'Elia[5] and Surya Saha[3,6,*]

1 Division of Biology, Kansas State University, Manhattan, KS 66506, USA
2 Allen County Community College, Burlingame, KS 66413, USA
3 Boyce Thompson Institute, Ithaca, NY 14853, USA
4 USDA-ARS, U.S. Horticultural Research Laboratory, Fort Pierce, FL 34945, USA
5 Indian River State College, Fort Pierce, FL 34981, USA
6 Animal and Comparative Biomedical Sciences, University of Arizona, Tucson, AZ 85721, USA

## ABSTRACT

Insects have a segmented body plan that is established during embryogenesis when the anterior–posterior (A–P) axis is divided into repeated units by a cascade of gene expression. The cascade is initiated by protein gradients created by translation of maternally provided mRNAs, localized at the anterior and posterior poles of the embryo. Combinations of these proteins activate specific gap genes to divide the embryo into distinct regions along the anterior–posterior axis. Gap genes then activate pair-rule genes, which are usually expressed in parts of every other segment. The pair-rule genes, in turn, activate expression of segment polarity genes in a portion of each segment. The segmentation genes are generally conserved among insects, although there is considerable variation in how they are deployed. We annotated 25 segmentation gene homologs in the Asian citrus psyllid, *Diaphorina citri*. Most of the genes expected to be present in *D. citri* based on their phylogenetic distribution in other insects were identified and annotated. Two exceptions were *eagle* and *invected*, which are present in at least some hemipterans, but were not found in *D. citri*. Many of the segmentation pathway genes are likely to be essential for *D. citri* development, and thus they may be useful targets for gene-based pest control methods.

**Submitted:** 24 December 2020

* Corresponding author. E-mail: suryasaha@cornell.edu

Preprint submitted at https://doi.org/10.1101/2020.12.24.424320

**Subjects** Genetics and Genomics, Animal Genetics, Bioinformatics

# DATA DESCRIPTION

## Introduction

Segmentation is the process by which repeated units of similar groups of cells are created along the anterior–posterior axis of a developing embryo. The molecular mechanisms involved in this process were first elucidated by large-scale developmental mutant screens in the insect model *Drosophila melanogaster* [1–5]. In *Drosophila*, segmentation begins with cytoplasmic inheritance of mRNAs that are maternally produced and provided to the oocyte. The products of these maternal-effect genes create gradients that define positional information within the embryo and activate a group of genes known as gap genes.

Gap genes are expressed in broad, well-defined domains in the early embryo and activate the next set of transcription factors: the pair-rule genes. Pair-rule genes are expressed in every other segment of the developing embryo. Together, they activate the

Annotating genes in *Diaphorina citri* genome version 3

Teresa D Shippy[1], S Miller[2], C Massimino[3], C Vosburg [Indian River State College[4], PS Hosmani[5], M Flores-Gonzalez[5], LA Mueller[5], WB Hunter[6], JB Benoit[7], SJ Brown[1], T D'elia[3], Surya Saha[5]

[1]Kansas State University; [2]Kansas State University, Allen County Community College; [3]Indian River State College; [4]Indian River State College, The Pennsylvania State University; [5]Boyce Thompson Institute; [6]USDA-ARS U.S. Horticultural Research Laboratory; [7]University of Cincinnati

Dec 17, 2020

Run

☆ Bookmark

⑂ Copy / Fork

2  Works for me    dx.doi.org/10.17504/protocols.io.bniimcce

D. citri annotation

Teresa Shippy
Kansas State University

**Figure 1.** Protocol for psyllid genome community curation [14]. https://www.protocols.io/widgets/doi?uri=dx.doi.org/10.17504/protocols.io.bniimcce

expression of segment polarity genes, which are expressed in every segment of the developing embryo. Comparative studies in diverse arthropod species have shown that some aspects of the segmentation pathway are highly conserved, while other aspects have undergone evolutionary change [6]. The hemipteran insects that have been examined seem to employ a particularly divergent method of segmentation. Most strikingly, the pair-rule genes, which are usually considered to be the most conserved portion of the segmentation pathway among insects, have lost their pair-rule expression and function and are expressed segmentally in at least some hemipterans [7–9].

## Context

As part of a community genome annotation project, we have annotated 25 homologs of *Drosophila* segmentation pathway genes in the genome of the hemipteran agricultural pest, *Diaphorina citri* (NCBI:txid121845), also known as the Asian citrus psyllid. A few segmentation genes were identified in a previous version of the *D. citri* genome [10]. However, many of those genes were incomplete because of genome assembly errors. The current genome version (v3) is much higher quality [11], allowing us to annotate genes with much higher accuracy and confidence. Except for *eagle* and *invected*, which appear to be missing from the *D. citri* v3 genome, we annotated all of the segmentation genes expected to be present in *D. citri*. Our annotations pave the way for future work aimed at understanding the expression and function of these genes during *D. citri* segmentation and the identification of essential genes that could be used as insect control targets.

## METHODS

Segmentation gene orthologs in *D. citri* genome v3 [11] were identified by BLAST (RRID:SCR_004870) search and confirmed by reciprocal BLAST. Manual annotation was performed in Apollo 2.1.0 (RRID:SCR_001936) [12] using available evidence, such as RNA-Seq reads, Iso-Seq transcripts and *de novo*-assembled transcripts. Multiple alignment and phylogenetic analysis were done in MEGA X (RRID:SCR_000667) [13]. More details of the annotation process are described in a protocols.io protocol (Figure 1) [14].

**Table 1.** Annotated *D. citri* segmentation gene orthologs.

| Group | Gene/Isoform | OGSv3 ID | Complete model | MCOT | Iso-Seq | RNA-Seq | Ortholog |
|---|---|---|---|---|---|---|---|
| Maternal Effect | | | | | | | |
| | *caudal* | Dcitr06g04620.1.4 | X | MCOT08625.0.OO | X | | |
| | *dorsal* | Dcitr02g07710.1.1 | X | MCOT04961.1.CO | X | X | X |
| | | Dcitr02g07710.1.2 | | MCOT04961.3.CO | | | |
| | *nanos* | Dcitr00g11810.1.1 | | MCOT03924.0.CT | | | X |
| | *TGFalpha* | Dcitr06g02470.1.1 | X | MCOT20343.1.CO | X | | |
| Gap | | | | | | | |
| | *tailless* | Dcitr01g16840.1.1 | X | | | | X |
| | *knirps related 1* | Dcitr10g03830.1.1 | X | | X | X | X |
| | *knirps related 2* | Dcitr10g03860.1.1 | X | MCOT03238.1.CT | | | X |
| | *knirps related 3* | Dcitr10g03875.1.1 | X | | | X | X |
| | *Kruppel* [†] | Dcitr00g04150.1.1 | | MCOT03100.0.CO | | X | X |
| | | Dcitr00g04150.1.2 | | | | | |
| | *hunchback* | Dcitr09g01780.1.1 | X | | X | X | |
| | | Dcitr09g01780.1.2 | | | | | |
| | *huckebein* | Dcitr04g11170.1.1 | X | MCOT23091.2.CT | | X | |
| | *orthodenticle* | Dcitr04g16960.1.1 | X | MCOT05861.1.CO | | X | X |
| | *empty spiracles* | Dcitr09g06330.1.1 | X | MCOT04329.0.CO | | | |
| | *cap-n-collar* | Dcitr03g12850.1.1 | X | MCOT01547.1.CT | | | |
| | *collier* | Dcitr03g01400.1.1 | X | MCOT16351.0.CO | | X | |
| | | Dcitr03g01400.1.2 | | MCOT09939.0.OO | | | |
| Pair-rule | | | | | | | |
| | *paired* | Dcitr01g09360.1.1 | X | MCOT01278.2.CO | | | X |
| | *odd skipped* | Dcitr01g20150.1.1 | X | MCOT15683.0.CO | | | |
| | *sloppy paired* | Dcitr02g08120.1.1 | X | MCOT19895.0.CT | | | |
| | *runt* | Dcitr01g07300.1.1 | X | MCOT14329.0.CT | | | X |
| | *even skipped* | Dcitr08g10250.1.1 | X | MCOT05103.0.CT | X | | |
| | *hairy* | Dcitr02g06890.1.1 | X | MCOT00149.0.CT | X | | X |
| | *odd paired* | Dcitr09g07980.1.1 | X | MCOT12397.2.CT | | | |
| | | | | MCOT03041.1.CT | | | |
| Segment polarity | | | | | | | |
| | *gooseberry* | Dcitr01g18500.1.1 | X | MCOT03447.1.CO | X | X | |
| | *engrailed* | Dcitr08g03480.1.1 | X | MCOT21494.0.CO | | X | |
| | | Dcitr08g03480.1.2 | | MCOT01333.0.CT | | | |
| Other related genes | | | | | | | |
| | *Runt related A* | Dcitr01g07290.1.1 | X | MCOT01714.2.CT | | | X |
| | *Runt related B* | Dcitr01g07260.1.1 | X | MCOT22437.2.CT | | | X |
| | *lozenge* | Dcitr01g07310.1.1 | | | X | | X |
| | *sister of odd and bowl* | Dcitr01g20160.1.1 | X | MCOT11314.0.MO | | | |
| | *brother of odd with entrails limited* | Dcitr01g11160.1.1 | X | | X | | |
| | *gooseberry neuro* | Dcitr01g18520.1.1 | X | MCOT03447.2.CT | X | X | X |

Each manually annotated gene has been assigned an OGSv3 gene identifier. Genes marked as complete models have a complete coding region. Evidence used for manual annotation was also recorded. Maker, Cufflinks, Oases Trinity (MCOT) [10] and Iso-Seq [11] indicate use of these genome-independent transcriptomes to validate the annotation. RNA-Seq indicates use of mapped RNA-Seq reads, while orthologs indicate comparison with related proteins was essential for the annotation. Additional information on specific evidence types is included in [14]. [†] denotes genes that were complete in OGSv2 but are partial in OGSv3.

## RESULTS AND DISCUSSION

We searched the *D. citri* v3 genome for orthologs of genes known to be involved in segmentation in *Drosophila* [15] (Table 1). We then used available evidence to manually annotate the genes that were present [11, 14] (Tables 1, 2). Most manual annotations were straightforward and performed using our workflow (see Methods), so only those requiring additional explanation are described in detail here.



**Table 2.** Segmentation gene ortholog number.

| | *Drosophila melanogaster* | *Apis mellifera* | *Tribolium castaneum* | *Acyrthosiphon pisum* | *Diaphorina citri* |
|---|---|---|---|---|---|
| **Maternal Effect** | | | | | |
| *caudal* | 1 | 1 | 1 | 1 | 1 |
| *dorsal* family (*dorsal, Dorsal-related immunity factor*) | 2 | 1 | 2 | 2 | 1 |
| *nanos* | 1 | 1 | 1 | 4 | 1 |
| *bicoid* | 1 | 0 | 0 | 0 | 0 |
| *oskar* | 1 | 0 | 0 | 0 | 0 |
| TGFa ligand (*gurken, Keren, spitz*) | 3 | 1 | 1 | 1 | 1 |
| **Gap** | | | | | |
| *tailless* | 1 | 1 | 1 | 1 | 1 |
| *knirps* family (*knirps, knirps-like, eagle*) | 3 | 3 | 2 | 3 | 3 |
| *giant* | 1 | 1 | 1 | 0 | 0 |
| *Kruppel* | 1 | 1 | 1 | 1 | 1 |
| *hunchback* | 1 | 1 | 1 | 1 | 1 |
| *huckebein* | 1 | 1 | 1 | 0 | 1 |
| *orthodenticle* | 1 | 2 | 2 | 1 | 1 |
| *buttonhead* | 1 | 1 | 1 | 0 | 0 |
| *empty spiracles* | 1 | 1 | 1 | 1 | 1 |
| *cap-n-collar* | 1 | 1 | 1 | 1 | 1 |
| *collier* | 1 | 1 | 1 | 1 | 1 |
| **Pair-rule** | | | | | |
| *paired* | 1 | 1 | 1 | 1 | 1 |
| *odd skipped* | 1 | 1 | 1 | 1 | 1 |
| *sloppy paired 1/sloppy paired 2* | 2 | 1 | 1 | 1 | 1 |
| *runt* | 1 | 1 | 1 | 1 | 1 |
| *even skipped* | 1 | 1 | 1 | 1 | 1 |
| *hairy* | 1 | 1 | 1 | 1 | 1 |
| *odd paired* | 1 | 1 | 1 | 1 | 1 |
| **Segment polarity** | | | | | |
| *gooseberry* | 1 | 1 | 1 | 1 | 1 |
| *engrailed* family (*engrailed, invected*) | 2 | 2 | 2 | 2 | 1 |

The *Drosophila melanogaster* numbers were determined from Flybase (RRID:SCR_006549) [16]. Ortholog numbers for *Apis mellifera* [17], *Tribolium castaneum* [18] and *Acyrthosiphon pisum* [19] are based on genome publications or NCBI (RRID:SCR_006472). *Diaphorina citri* ortholog numbers represent our final manual annotation.

## Maternal effect genes

One-to-one orthologs of *caudal* (*cad*), *dorsal* (*dl*), and *nanos* (*nos*) were found in the *D. citri* v3 genome. *dl* was previously annotated in the *D. citri* genome v1.1 because of its role in innate immunity [10]. Here, we annotated a second isoform of *dl* (Table 1).

We also identified a single TGF-α ligand-encoding gene in *D. citri*. In *Drosophila*, there are three *TGF-α* ligand paralogs (*grk, Krn, spi*), but in many other insects, only one *TGF-α* ligand has been identified [17–21]. Orthologs for *bicoid* (*bcd*) and *oskar* (*osk*) were not found in *D. citri* (Table 2), which is consistent with the previously described phylogenetic distribution of these genes [22].

## Gap genes

One-to-one orthologs of the gap genes *tailless* (*tll*), *Kruppel* (*Kr*), *hunchback* (*hb*), *huckebein* (*hkb*), *empty spiracles* (*ems*), *cap-n-collar* (*cnc*), and *collier* (*col*) were identified and annotated in the *D. citri* v3 genome (Tables 1, 2). Except for *hkb*, which was reported

missing in *Acyrthosiphon pisum* (*A. pisum*) [19], conservation of these genes was expected (Table 2). Two *knirps-related* (*knrl*) genes and two *orthodenticle* (*otd*) homologs were annotated and are discussed in more detail below. *giant* (*gt*) and *buttonhead* (*btd*) appear to be absent in the v3 *D. citri* genome assembly (Table 2).

Phylogenetics of the *knirps* family suggests that a single ancestral gene duplicated early in the insect lineage, producing two paralogs called *knirps-related* (*knrl)* and *eagle* (*eg*) [23]. Subsequent duplications have occurred in various insect lineages. A duplication in the lineage leading to *Drosophila* resulted in the paralogs *knirps* and *knirps-like* (also called *knirps-related*) [23]. A separate duplication of *knrl* seems to have occurred in the hemipteroid lineage, leading to three *knirps* family genes (two *knrl* and one *eg*) in most hemipterans [23]. In the *D. citri* genome v3 we identified three potential *knirps* family genes (Tables 1, 2), one of which was annotated as *knirps* in *D. citri* genome v1.1 [10]. These three *knirps* family genes are all located on the same chromosome, within a 900-kilobase pair (Kb) region. All three predicted proteins contain the highly conserved 94-amino acid N-terminal domain and the C-terminal PIDLS motif commonly found in *knirps* family members [23]. However, none of them contain the GASS-domain motif that is unique to the Eg protein [23]. Owing to the lack of this signature Eagle motif, the resulting *D. citri* annotations were named *knirps-related 1* (*knrl1*), *knirps-related 2* (*knrl2*) and *knirps-related 3* (*knrl3*). Despite the lack of GASS domain, it is possible that one of these genes is the ortholog of *eg* but has lost the characteristic motif. Interestingly, *D. citri knrl1* has a small exon just 5′ of the highly conserved coding exon that is the first exon in most *knirps* family genes (Figure 2). Similar gene structure has been reported for one *knirps* family gene each in *D. melanogaster*, the honeybee *Apis mellifera*, *A. pisum*, and the human louse *Pediculus* [23]. All but *D. melanogaster* share small stretches of sequence identity in the amino acid sequence encoded by this additional exon, suggesting that the 5′ exon might have been present in a common ancestor. This model suggests the duplication and acquisition of an additional exon by one paralog early in the insect lineage. The paralog containing the additional exon appears to have been lost in holometabolous insects sometime after the divergence of the Hymenoptera from the rest of the Holometabola.

Most insects have two *otd* genes. However, in both *Drosophila* and *A. pisum* only one *otd* gene has been identified. *Drosophila* is missing the *otd-2* ortholog, while *otd-1* has apparently been lost in pea aphids [24]. In the *D. citri* v3 genome, we found two *otd* genes adjacent to one another on chromosome 4 (Table 1). Phylogenetic analysis suggests that one of these genes is an *otd-1* ortholog, while the other is an *otd-2* ortholog (Figure 3). The genomic clustering of *otd-1* and *otd-2* has also been noted in other insects and crustaceans where their genomic location has been examined [25], providing further support for the identification of the *D. citri* genes as *otd-1* and *otd-2*. The presence of *otd-1* in *D. citri* suggests that *otd-1* was lost in the pea aphid lineage after divergence of the psyllid and aphid lineages.

*gt* is conserved in some, but not all, hemipterans. No *gt* ortholog was found in the *A. pisum* genome [19]; however, apparent *gt* orthologs are present in *Rhodnius prolixus, Cimex lectularius, Halyomorpha halys, Oncopeltus fasciatus* and *Bemisia tabaci* [26–30]. RNA interference (RNAi) studies indicate that the *R. prolixus* and *O. fasciatus gt* orthologs both function as gap genes [26, 29]. We performed BLAST searches of *D. citri* genome v3 with all of these hemipteran *gt* orthologs, but were unable to identify a *D. citri gt* ortholog (Table 2).

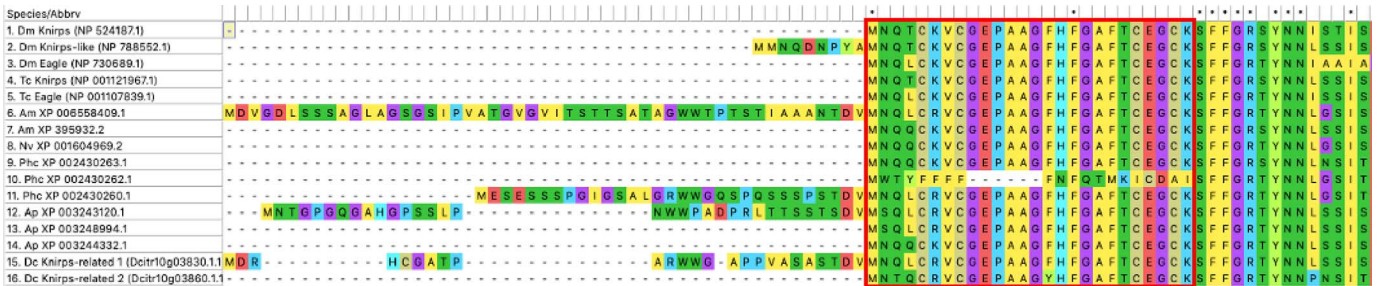

**Figure 2.** Multiple alignment of the N-terminal region of Knirps family proteins. The red box denotes the sequence encoded by a highly conserved exon [23]. Species represented are *Drosophila melanogaster* (Dm), *Tribolium castaneum* (Tc), *Apis mellifera* (Am), *Nasonia vitripennis* (Nv), *Pediculus humanus* (Ph), *Acyrthosiphon pisum* (Ap) and *Diaphorina citri* (Dc). Named proteins have been manually annotated, while those with only an accession number are computationally predicted. Five of the *knirps* family genes (one each in Dm, Am, Ph, Ap and Dc) have sequence upstream of the universally conserved sequence that typically begins in the first exon [23, and this work]. In these proteins the conserved core sequence (red box) begins in the second exon. Four of the five proteins with an additional 5′ exon (Am, Ph, Ap and Dc) share a small region of sequence identity at the C-terminal end of the sequence encoded by this exon.

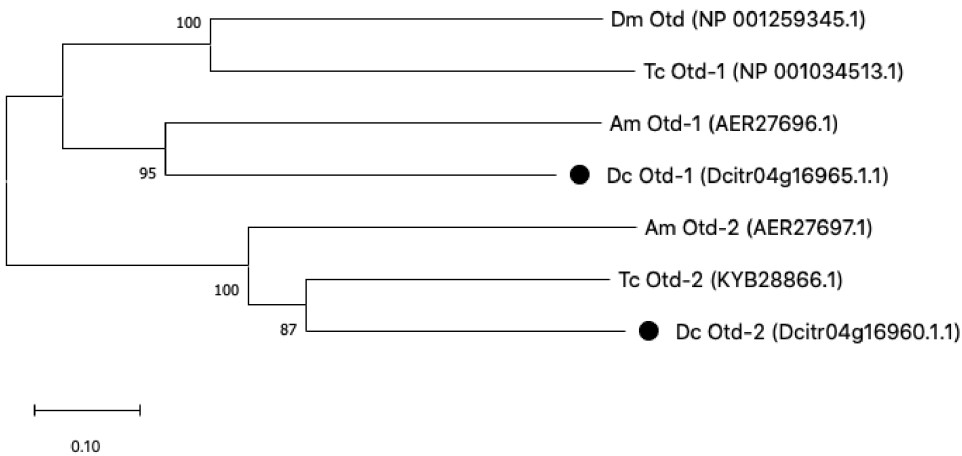

**Figure 3.** Neighbor joining tree of Otd homologs from *Drosophila melanogaster* (Dm), *Tribolium castaneum* (Tc), *Apis mellifera* (Am), and *Diaphorina citri* (Dc). *D. citri* proteins are marked with black circles.

*btd* is a member of the Sp family of transcription factors. Recent reports indicate that the ancestral state for arthropods—and perhaps all metazoans—is likely to be the presence of three Sp members [31]. These three Sp family genes cluster into three monophyletic clades (Sp5/btd, Sp1-4/(Sp-pps) and Sp6-9 (Sp1)) [31]. Even though the ancestral state appears to be the presence of three Sp family members, *btd* is absent from the *A. pisum* genome [19]. Furthermore, repeated efforts to clone *btd* from *Oncopeltus fasciatus* have only resulted in the identification of the two non-*btd* Sp genes. This suggests that *btd* may have been lost in the lineage leading to hemipterans. We were also unable to find a true *btd* ortholog in either the *D. citri* genome v3, or in independent *de novo* transcriptomes (Table 2). Two Sp family members were found that appear to be orthologous to *Sp1* and *Spps*, but these were not annotated.

## Pair-rule genes

One-to-one orthologs were found for all pair-rule genes examined, including *paired* (*prd*), *odd skipped* (*odd*), *sloppy paired* (*slp*), *runt* (*run*), *even skipped* (*eve*), *hairy* (*h*) and *odd paired*

(*opa*) in the *D. citri* genome v3 (Tables 1, 2). A partial copy of *h* had been annotated in a previous genome version [10]. Three of the pair-rule genes we annotated have closely related paralogs and required additional analysis before gene identities could be assigned.

Prd is a member of the Pax3/7 family of proteins. In *Drosophila* there are three Pax3/7 family genes, which are known to be involved in segmentation and neurogenesis: *prd*, *gooseberry* (*gsb*) and *gooseberry-neuro* (*gsb-n*). While the number of Pax3/7 genes varies in arthropods, data from insects and arachnids suggest that the roles of Pax3/7 in segmentation and neurogenesis are likely to be conserved in all arthropods [32]. In the *D. citri* genome v3, we also found three Pax3/7 genes, which we named *paired*, *gooseberry* and *gooseberry-neuro* based on reciprocal BLAST analysis and genomic location (Tables 1, 2). The *gsb* and *gsb-n* orthologs are discussed in more detail in the segment polarity gene section.

*odd* is a zinc finger transcription factor with three close relatives known as *brother of odd with entrails limited* (*bowl*), *sister of odd and bowl* (*sob*) and *drumstick* (*drum*). All four genes are in a conserved cluster in *Tribolium castaneum* and *D. melanogaster* [33]. In *D. citri* genome v3, *drum, odd* and *sob* are all located within a 400-Kb region, with *odd* and *sob* overlapping one another on opposite strands. It remains unclear whether the overlap is correct or results from misassembly, but the genes are almost certainly located very close together. *D. citri bowl* is located on the same chromosome about 20 megabase pairs (Mb) away. Separation of *bowl* from the rest of the cluster has also been observed in *Anopheles gambiae* [34].

Insects have four *runt* domain-containing genes: *run, Runt-related A (RunxA)*, *Runt-related B (RunxB)* and *lozenge (lz)*. All four genes are typically found in a cluster and their order and orientation is well conserved across insects [35] (Figure 4). We were able to annotate full length models for all four genes in the *D. citri* genome. It appears that the cluster is intact, with all four genes identified in their expected order within a 300-Kb region (Figure 4).

## Segment polarity genes

Many segment polarity genes are members of the Wnt and Hedgehog signaling pathways. Manual annotation of the Wnt pathway genes in the *D. citri* genome v3 is described in a separate report [36]. Here, we report the manual annotation of the segment polarity genes *gooseberry* (*gsb*) and *engrailed* (*en*) (Table 1). *gsb* and *en* each have a tightly linked paralog in many insects [37–39]. Surprisingly, we were unable to find the *en* paralog *invected* (*inv*) in the current genome assembly or the *de novo* transcriptome. However, we did find and annotate the *gsb* paralog *gooseberry-neuro* (*gsb-n*) in its expected position adjacent to *gsb* (Table 1). This positional information helped verify the identity of *gsb-n*, since phylogenetic analysis was inconclusive.

## CONCLUSION

We searched for orthologs of 33 *Drosophila* segmentation genes in the *D. citri* v3 genome and identified and annotated 25 homologous genes. We were unable to find orthologs for 10 of the *Drosophila* genes, while *D. citri* has one segmentation gene (*otd-2*) whose ortholog has been lost in *Drosophila*. Most of the absences, except *eagle* and *invected*, were expected, based on the known phylogenetic distribution of the genes. While all the genes discussed in this report were initially identified because of their role in embryonic patterning and



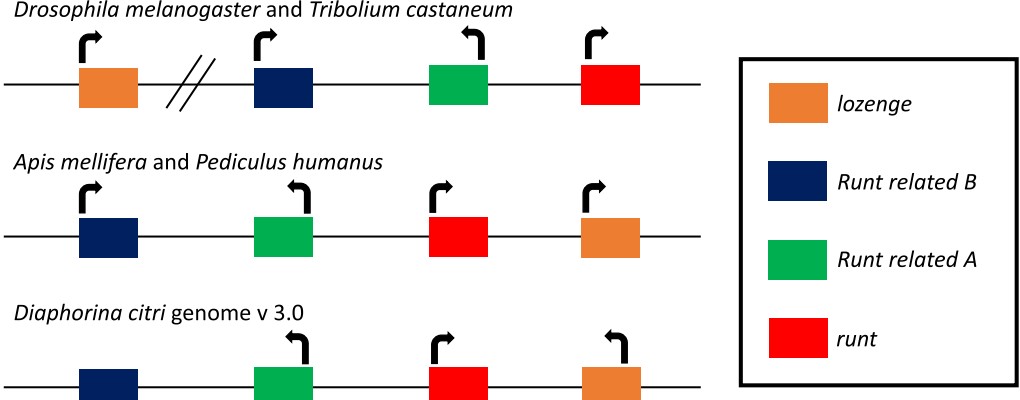

**Figure 4.** Runt domain (RD) cluster in representative insects. Cluster information from other insects was obtained from [35]. The RD clusters in *D. melanogaster* and *T. castaneum* have three genes in a core cluster, with *lozenge* (*lz*) separated from the cluster, but on the same chromosome distal to *runt* (*run*). The RD clusters in *A. mellifera* and *P. humanus* have all four genes clustered together with *lz* proximal to *run*. The RD cluster in *D. citri* most closely follows the pattern seen in *A. mellifera* and *P. humanus*. *D. citri lz* appears to be transcribed in the opposite direction compared to other insects, but it is possible that this is attributed to local misassembly. The orientation of *D. citri Runt-related B* (*RunxB*) is uncertain, since there are tandem artifactual duplicates that are on opposite strands. We chose the *RunxB* copy closest to *Runt-related A* (*RunxA*) for annotation. Future assembly improvements may help resolve the gene orientation in this cluster.

segmentation in *Drosophila*, many have other important functions, such as pole cell development, neural stem cell maintenance, sex determination, and immune function. Analysis of expression patterns and gene function will be required to determine which of these genes are involved in *D. citri* segmentation.

## DATA VALIDATION AND QUALITY CONTROL

Orthology of annotated genes was verified by reciprocal blasts and phylogenetic analysis. Completeness of gene models (defined as a complete coding region) was assessed by comparing the encoded protein to orthologous proteins. Evidence used for validation of gene models is shown in Table 1.

## REUSE POTENTIAL

This manual curation was carried out as a part of the *D. citri* community annotation project [40], with a goal to annotate gene families related to immune response, metabolism and other major functions [36, 41–43]. As scientists search for ways to control the spread of Huanglongbing, understanding the development pathways of its vector, *D. citri*, may provide insights into essential genes that could be targeted by pest control methods. The availability of accurate gene models will facilitate the design of experiments aimed at understanding the expression and function of these genes.

## DATA AVAILABILITY

The gene models will be part of an updated official gene set (OGS) for *D. citri* that will be submitted to NCBI. The OGS (v3) will also be publicly available for download, BLAST analysis and expression profiling on Citrusgreening.org and the Citrus Greening Expression Network [40]. The *D. citri* genome assembly (v3), OGS (v3) and transcriptomes are accessible on Citrusgreening.org and NCBI. Accession numbers for genes used in multiple

**Table 3.** Orthologs used in alignments and phylogenetic analysis.

| Species | Accession | Name in NCBI | Name in Tree |
|---|---|---|---|
| *Drosophila melanogaster* | NP_524187.1 | knirps, isoform A | Dm Knirps |
| *Drosophila melanogaster* | NP_788552.1 | knirps-like, isoform A | Dm Knirps-like |
| *Drosophila melanogaster* | NP_730689.1 | eagle, isoform B | Dm Eagle |
| *Tribolium castaneum* | NP_001121967.1 | knirps | Tc Knirps |
| *Tribolium castaneum* | NP_001107839.1 | eagle | Tc Eagle |
| *Apis mellifera* | XP_006558409.1 | nuclear receptor subfamily 1 group D member 2 isoform X1 | Am XP_006558409.1 |
| *Apis mellifera* | XP_006559124.1 | knirps-related protein | Am XP_395932.2 |
| *Nasonia vitripennis* | XP_001604969.2 | protein embryonic gonad | Nv XP_001604969.2 |
| *Pediculus humanus corporis* | XP_002430263.1 | conserved hypothetical protein | Phc XP_002430263.1 |
| *Pediculus humanus corporis* | XP_002430262.1 | conserved hypothetical protein | Phc XP_002430262.1 |
| *Pediculus humanus corporis* | XP_002430260.1 | knirps related protein, putative | Phc XP_002430260.1 |
| *Acyrthosiphon pisum* | XP_003243120.1 | knirps-related protein-like | Ap XP_003243120.1 |
| *Acyrthosiphon pisum* | XP_003248994.1 | protein embryonic gonad-like | Ap XP_003248994.1 |
| *Acyrthosiphon pisum* | XP_003244332.1 | protein doublesex* | Ap XP_003244332.1 |
| *Drosophila melanogaster* | NP_001259345.1 | ocelliless, isoform G | Dm Otd |
| *Tribolium castaneum* | NP_001034513.1 | orthodenticle-1 | Tc Otd-1 |
| *Tribolium castaneum* | KYB28866.1 | orthodenticle-2 | Tc Otd-2 |
| *Apis mellifera* | AER27696.1 | orthodenticle 1 | Am Otd-1 |
| *Apis mellifera* | AER27697.1 | orthodenticle 2 | Am Otd-2 |

Species, accession number, full name and abbreviated name are provided for all orthologs used in multiple alignments and phylogenetic trees [18, 21, 23, 45–50]. The asterisk (*) denotes a gene that appears to be incorrectly named *doublesex* in GenBank, but is actually a *knirps* family gene.

alignments or phylogenetic trees are provided in Table 3, and all additional data is available via the *GigaScience* GigaDB repository [44].

## EDITOR'S NOTE

This article is one of a series of Data Releases crediting the outputs of a student-focused and community-driven manual annotation project curating gene models and, if required, correcting assembly anomalies, for the *Diaphorina citri* genome project [11].

## DECLARATIONS
## LIST OF ABBREVIATIONS

Am: *Apis mellifera*; Ap: *Acyrthosiphon pisum;* CGEN: Citrus Greening Expression Network; Dc: *Diaphorina citri*; Dm: *Drosophila melanogaster*; Kb: kilobase pair; MCOT: Maker, Cufflinks, Oases and Trinity; NCBI: National Center for Biotechology Information; Nv: *Nasonia vitripennis*; OGS: official gene set; Ph: *Pediculus humanus*; RD: Runt domain; RNA-Seq: RNA sequencing; Tc: *Tribolium castaneum*.

## ETHICAL APPROVAL

Not applicable.

## CONSENT FOR PUBLICATION

Not applicable.

## COMPETING INTERESTS

The authors declare that they have no competing interests.

## FUNDING

This work was supported by USDA-NIFA grant 2015- 70016-23028, HSI 1300394, 2020-70029-33199 and an Institutional Development Award (IDeA) from the National Institute of General Medical Sciences of the National Institutes of Health under grant number P20GM103418.

## AUTHOR CONTRIBUTIONS

WBH, SJB, TD and LAM conceptualized the study; TDS, SJB, TD and SS supervised the study; SJB, TD, SS and LAM were involved in project administration; SM and TDS conducted the investigation; PH, MF-G and SS contributed to software development; TDS, SS, PH and MF-G developed the methodology; SJB, TD, WBH and LAM acquired funding; SM and TDS wrote the original draft; SJB, WBH and SS reviewed and edited the draft.

## ACKNOWLEDGEMENTS

We thank Will Tank for technical assistance and helpful discussions.

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
