## [Reviewer Report]

Reviewer name and names of any other individual's who aided in reviewer Mary Ann TuliDo you understand and agree to our policy of having open and named reviews, and having your review included with the published papers. (If no, please inform the editor that you cannot review this manuscript.)YesIs the language of sufficient quality?YesPlease add additional comments on language quality to clarify if needed
The manuscript reads very well.Are all data available and do they match the descriptions in the paper? NoAdditional Comments1) Line 206. "Multiple alignments were performed with MUSCLE or MEGA7 " (figure 1).
We need the output of MUSCLE (FASTA).
We need the output of MEGA7 (FASTA)

2) I note that MEGA7 has been used. I wonder why the newer release (MEGAX, March '21) was not used. Furthermore, the annotation protocol (dx.doi.org/10.17504/protocols.io.bniimcce) suggests using Mega7 or MegaX.

3) Line 207. "phylogenetic analysis was done in MEGA7 or MEGA X" (figure 2).
We need the files underlying the phylogenetic tree (newick) (figure 2).
Are the data and metadata consistent with relevant minimum information or reporting standards? See GigaDB checklists for examples <a href="http://gigadb.org/site/guide" target="_blank">http://gigadb.org/site/guide</a>YesAdditional CommentsNomenclature standards have been met.
All cited INSDC accession numbers are publicly available.Is the data acquisition clear, complete and methodologically sound?YesAdditional CommentsCuration workflow used for community annotation is available via protocols.io , nonetheless the manuscript includes comprehensive summary which is appropriate.Is there sufficient detail in the methods and data-processing steps to allow reproduction?NoAdditional CommentsSee "Are all data available and do they match the descriptions in the paper?" above.
Once the additional files are made available I believe reproduction will be possible.Is there sufficient data validation and statistical analyses of data quality? YesAdditional CommentsIs the validation suitable for this type of data?YesAdditional CommentsIs there sufficient information for others to reuse this dataset or integrate it with other data?YesAdditional CommentsAny Additional Overall Comments to the AuthorSome of my comments/recommendations are pertinent to the other D. citri manuscripts currently under review.

Please send the files to us via the FTP server details at the end of this email.
Please note the following.

We strongly recommend filenames are concise and meaningful (e.g. "gene_exp_cell_lines.csv" is better than "Supp1.txt").
Filenames should be unique.
Filenames should not include spaces. We recommend using the underscore (_) in place of spaces in the filenames.
Filenames should only include the following characters a-z,A-Z,0-9,_,-,+,.
Text and tabular information must be in a non-proprietary and text-based format (e.g. CSV/TSV rather than PDF or XLS)
Images must be in a lossless raster format (e.g. TIFF, PNG) or vector format (e.g. SVG).
If you encounter any errors please send us the full context and error message to help us resolve the problem.

username = user11
password = SahacitriSegm
FTP server = parrot.genomics.cn

For using tools like FileZilla use the standard FTP protocol (not sftp).

If you are using command line FTP, you may need to use the passive mode (e.g. use epsv command):

> ftp @parrot.genomics.cn
Connected to parrot.genomics.cn.
220 (vsFTPd 2.0.5)
331 Please specify the password.
Password:
230 Login successful.
Remote system type is UNIX.
Using binary mode to transfer files.
ftp> epsv
EPSV/EPRT on IPv4 off.
ftp> put
local: remote:
227 Entering Passive Mode (218,188,108,76,126,77)
150 Ok to send data.
100% |************************************************************************| 12 137.86 KiB/s 00:00 ETA
226 File receive OK.
12 bytes sent in 00:00 (0.26 KiB/s)
ftp>RecommendationMinor Revision

---

## [Reviewer Report]

Reviewer name and names of any other individual's who aided in reviewer Hailin LiuDo you understand and agree to our policy of having open and named reviews, and having your review included with the published papers. (If no, please inform the editor that you cannot review this manuscript.)YesIs the language of sufficient quality?YesPlease add additional comments on language quality to clarify if needed
Are all data available and do they match the descriptions in the paper? YesAdditional CommentsAre the data and metadata consistent with relevant minimum information or reporting standards? See GigaDB checklists for examples <a href="http://gigadb.org/site/guide" target="_blank">http://gigadb.org/site/guide</a>YesAdditional CommentsIs the data acquisition clear, complete and methodologically sound?YesAdditional CommentsIs there sufficient detail in the methods and data-processing steps to allow reproduction?YesAdditional CommentsIs there sufficient data validation and statistical analyses of data quality? YesAdditional CommentsIs the validation suitable for this type of data?YesAdditional CommentsIs there sufficient information for others to reuse this dataset or integrate it with other data?YesAdditional CommentsAny Additional Overall Comments to the AuthorIt seems there are no very sound biological values in this manuscript, and more validation or comparative study are suggested to mine more meaningful conclusions.RecommendationReject (Unsound or Unusuable)